# Ultra-compact snapshot spectral light-field imaging

Xia Hua[1,9], Yujie Wang[2,9], Shuming Wang [1,3,4,9 ✉], Xiujuan Zou[1], You Zhou [1], Lin Li [5,6], Feng Yan[1], Xun Cao[1,3,4 ✉], Shumin Xiao [2,7 ✉], Din Ping Tsai [5 ✉], Jiecai Han[8], Zhenlin Wang [1,4 ✉] & Shining Zhu [1,3,4 ✉]

Ideal imaging, which is constantly pursued, requires the collection of all kinds of optical information of the objects in view, such as three-dimensional spatial information (3D) including the planar distribution and depth, and the colors, i.e., spectral information (1D). Although three-dimensional spatial imaging and spectral imaging have individually evolved rapidly, their straightforward combination is a cumbersome system, severely hindering the practical applications of four-dimensional (4D) imaging. Here, we demonstrate the ultra-compact spectral light-field imaging (SLIM) by using a transversely dispersive metalens array and a monochrome imaging sensor. With only one snapshot, the SLIM presents advanced imaging with a 4 nm spectral resolution and near-diffraction-limit spatial resolution. Consequently, visually indistinguishable objects and materials can be discriminated through SLIM, which promotes significant progress towards ideal plenoptic imaging.

[1] National Laboratory of Solid State Microstructures, School of Physics, School of Electronic Science and Engineering, Nanjing University, Nanjing 210093, China. [2] Ministry of Industry and Information Technology Key Lab of Micro-Nano Optoelectronic Information System, Harbin Institute of Technology (Shenzhen), Shenzhen 518055, China. [3] Key Laboratory of Intelligent Optical Sensing and Manipulation, Ministry of Education, Nanjing University, Nanjing 210093, China. [4] Collaborative Innovation Center of Advanced Microstructures, Nanjing 210093, China. [5] Department of Electrical Engineering, City University of Hong Kong, Kowloon, Hong Kong, China. [6] State Key Laboratory of Precision Spectroscopy, School of Physics and Electronic Science, East China Normal University, Shanghai 200062, China. [7] Pengcheng Laboratory, Shenzhen 518055, China. [8] National Key Laboratory of Science and Technology on Advanced Composites in Special Environments, Harbin Institute of Technology, Harbin 150080, China. [9] These authors contributed equally: Xia Hua, Yujie Wang, Shuming Wang. ✉email: wangshuming@nju.edu.cn; caoxun@nju.edu.cn; shumin.xiao@hit.edu.cn; dptsai@cityu.edu.hk; dptsai@cityu.edu.hk; zhusn@nju.edu.cn

Optical imaging is an important technology widely used for collecting the spatial information of objects, from giant mountains and buildings to microscopic cells and even molecules. To address the insufficiency in the depth resolution of planar imaging, various three-dimensional (3D) imaging techniques, such as light-field imaging[1,2], stereo vision[3], structured light illumination[4], and time of flight methods[5] with additional light sources, have been employed to effectively obtain the 3D spatial information of the captured scene or objects. Furthermore, color imaging based on Maxwell's three-primary colors theory introduces a new dimension, i.e., the spectral dimension, to traditional monochrome imaging, which simply integrates all the spectra into a single intensity. Although tricolor mechanisms (red, green, blue) are extensively employed in commodity imaging and display products, the demand for full spectral information is becoming increasingly urgent in various application scenarios such as material discrimination, industrial inspection, and metamerism recognition. Hence, the integration of traditional imaging and spectroscopy has become an inevitable trend of optical imaging evolution. During the past decade, many efficient spectral imaging techniques combining traditional planar imaging have been developed, e.g., coded aperture snapshot spectral imager (CASSI)[6], computed tomographic imaging spectrometer (CTIS)[7], and prism-mask modulation imaging spectrometer (PMIS)[8]. Despite their impressive performances and snapshot capability, all kinds of optical elements embedded in cameras, such as prisms, lenses, gratings, and masks, are extremely cumbersome, which severely prevents cameras from being more widely applied. On the other hand, an advanced imaging technique that can acquire four-dimensional information (4D information: 3D spatial information plus 1D spectral information) with ultra-compact size and high-quality performance, has not yet been demonstrated.

Recently, metasurfaces have been desired for their compactness, which makes them promising alternatives to the heavy and complicated bulk optical devices[9,10]. A metasurface consisting of dense arrangements of nano-antennas could precisely control the phase, intensity, polarization, orbital angular momentum, and frequency of incident light[9–13]. To date, among all metasurface-based planar photonic devices, the metalens is the most fundamental and prominent[14,15]. By tailoring the nano-antennas, the ultrathin metalenses have shown equivalent or even better performances in terms of efficiency[16], the numerical aperture (NA)[17], broadband achromatism[18,19], coma cancellation[20], etc. Very recently, metalens-array-based light field imaging has also been demonstrated to obtain 3D information in the visible regime without any chromatic aberration[21]. Pioneering works have also utilized metasurfaces or other nanostructures to obtain high-quality spectra in compact configurations[22–25]. Nevertheless, though this progress is a good basis for spectral information acquisition, 4D imaging is still far off due to the difficulty in achieving high-quality spectra and 3D spatial resolution simultaneously. In this work, by taking advantage of transversely dispersive metalens arrays, we demonstrate ultra-compact spectral light-field imaging (SLIM) to record 4D information through a single snapshot using a monochrome sensor.

## Results

### Characteristics of transversely dispersive metalens array.
The main part of SLIM is a $48 \times 48$ TiO$_2$ based metalens array combined with a monochrome CMOS sensor. Because the metalens array could capture 3D spatial information, the key step towards 4D imaging is the capture of spectral information, which requires a transversely dispersive metalens. Basically, the focus of plane wave light at an arbitrary point in the focal plane $(x', y', f)$ needs a phase profile of the form:

$$\varphi(x,y,\lambda) = -\frac{2\pi}{\lambda}(\sqrt{(x-x')^2 + (y-y')^2 + f^2} \\ - \sqrt{(x')^2 + (y')^2 + f^2} - x\sin\theta) \quad (1)$$

where $f$ is the focal length of the metalens and $\theta$ is the incident angle. To realize transverse dispersion, the broadband incident beam must be focused onto a fixed focal plane with a large lateral shift at different wavelengths. Metalenses naturally have strong chromatic aberrations, with the focal spots moving along the propagational direction of light at different wavelengths. However, this transverse dispersion is also affected by out-of-focus blur, which is not helpful for the extraction of spectral information in a snapshot imaging system. In contrast, transverse chromatic aberration, with images of different wavelengths being spread out across the imaging plane, results in more differences between different wavelengths, which can greatly facilitate the extraction of spectral information. To realize this large phase compensation, nano-pillars and nano-holes with a high aspect ratio are employed. Figure 1a shows a schematic of the transversely dispersive imaging of a metalens. Here, the phase division principle is employed to address this dispersion manipulation case[18,19,26]. The compensation phase is required to control the phase profile provided by the metalens at different wavelengths. Taking the visible spectrum $\{\lambda_b, \lambda_r\}$ as an example, the compensation phase between red and blue light is expressed as $\Delta\varphi(x) = \varphi(x_b, 0, \lambda_b) - \varphi(x_r, 0, \lambda_r) + \phi_{shift}$, where $\phi_{shift} = \max|\varphi_r - \varphi_b|$ is introduced to make the designed nano-antennas able to provide the required compensation phase. This compensation phase can be provided by the waveguide resonance in the specifically designed nano-pillars and inversed structures (nanoholes)[19,21] (see Supplementary Note 1: Designs and numerical simulation for details). In this work, we choose the working band to be {400 nm, 667 nm}, and $x_b = 11.15\,\mu m$, $x_r = -13.85\,\mu m$, and $f = 165\,\mu m$. Under an incident angle $\theta = -16°$, optimal imaging with transverse dispersion can be achieved, with the phase profiles at different wavelengths shown in Fig. 1b. The numerical simulation of the transversely dispersive focusing with this metalens is shown in Fig. 1c, which fits the phase division design well.

Based on the above design, a TiO$_2$ metalens array has been experimentally fabricated via a combined process of electron-beam lithography and inductively coupled plasma (see Supplementary Note 2: Device fabrication for detailed information). In contrast to previous techniques with atomic layer deposition[16], a top-down etching process has been employed in current research to improve the mass-fabrication, production yield and time consumption of TiO$_2$ metalenses. Figure 1d shows a top-view scanning electron microscopy (SEM) image of a TiO$_2$ metalens array that consists of $48 \times 48$ metalenses. Each metalens has a diameter of $30\,\mu m$ and contains more than 25,000 TiO$_2$ nanopillars and nanoholes (see Fig. 1e, f). The tilt-view SEM image in Fig. 1g shows that the nano-pillars have nearly perfect vertical sidewalls, which is of crucial importance for efficiently controlling the effective refractive index at any position of the metalens array. Here, the maximal aspect ratio reaches 40, which is much larger than that of previous reports. Our fabrication method can greatly promote the performance of TiO$_2$ metalenses and expand their potential applications. The optical function of the metalens has been characterized, and the good performance proves the fine fabrication (see Supplementary Note 3: Optical measurements for details). Transverse dispersive imaging of the metalens has also been investigated. Using white light illumination with a transmission window from 450 to 650 nm, the letter "4" is imaged by the metalens to different positions at different

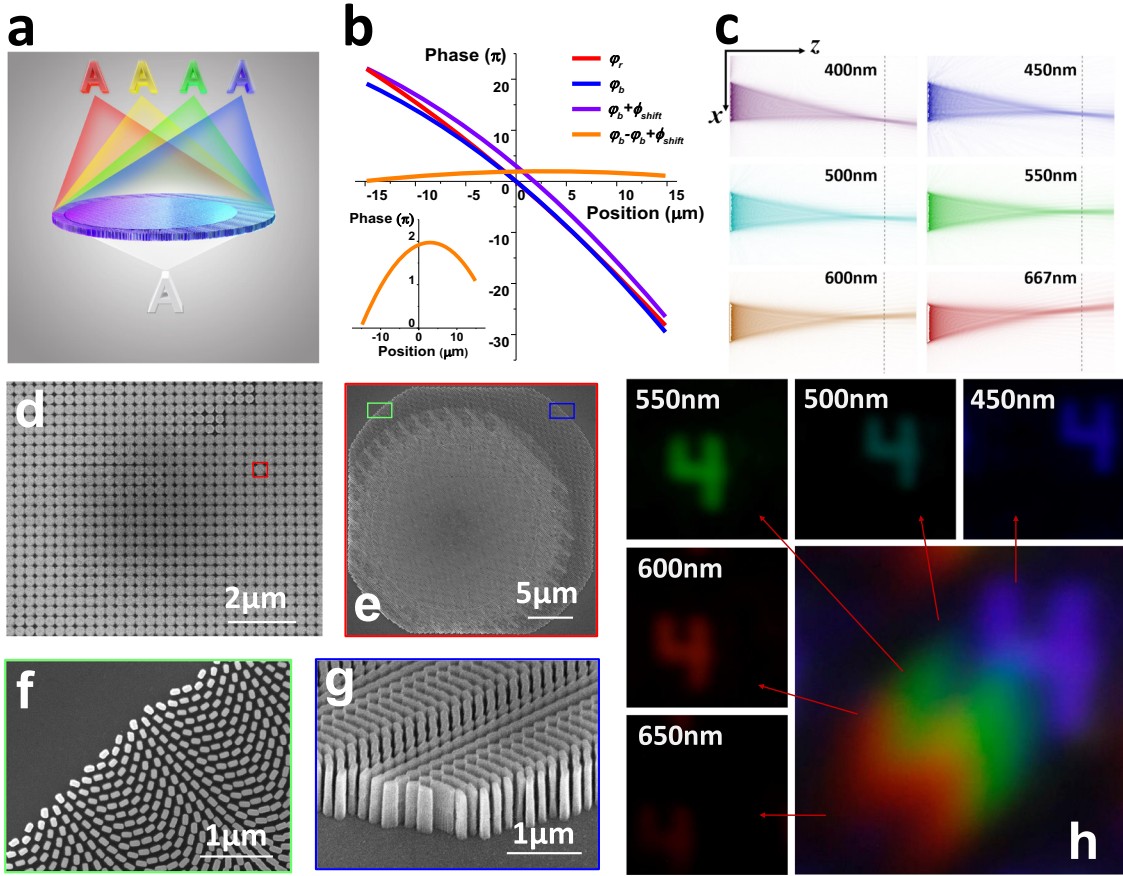

**Fig. 1 The details of transversely dispersive metalens array. a** Schematic of the transversely dispersive metalens. **b** Phase distributions at different wavelengths and phase compensation produced by the metalens. Inset shows the required phase compensation of the metalens. **c** Simulated focal spot intensity profile for different wavelengths. **d** Scanning electron microscope (SEM) image of the fabricated metalens array. **e** Top view of the red region in (**d**) (single metalens). Zoomed-in SEM image (**f**) of the green region in (**e**). Tilted view (**g**) of the blue region in (**e**). Image of a letter "4" by the metalens with a white light illumination with a transmission window of 450nm–650 nm is shown in (**h**).

wavelengths. The evident shift between images shows the transverse dispersion (see Fig. 1h). However, this also leads to a large overlap between the images of different wavelengths and consequently results in a blurred image.

**System design demonstration**. In contrast to utilizing extra optical elements such as coded mask[6,8] or auxiliary camera[27] to avoid overlap and blur, a spectrum reconstruction algorithm has been introduced into the SLIM system to obtain the spectral information at every position in a scene. Moreover, the whole area of the metalens is used to receive the incident light. Considering that the thickness and focal length of the metalens array are small, the SLIM camera can realize both the best light throughput and the highest system compactness. To clarify the entire process of SLIM system, the acquisition and reconstruction scheme of proposed SLIM system, conventional light field imaging system[2] and snapshot spectral imaging system[6] are shown in Fig. 2 and Supplementary Movie 1.

(1) As for the SLIM system shown in Fig. 2a, the 4D "$x + y + z + \lambda$" data cube is modulated by metalens array and decoupled into multi-view "$x + y + \lambda$" information. Then, through the high dispersion of the proposed metalens, blurred image (caused by the integration of spectral overlap) is formed behind each metalens and captured by the camera. Finally, utilizing the proposed light field spectrum reconstruction algorithm (an inherent convex

optimization method), both "$x + y + \lambda$" and "$x + y + z$" imaging result of the captured scene can be recovered.

(2) Figure 2b describes the classic light field imaging, which lacks the capability of encoding spectral information, layered depth images (LDI) can be obtained with the tradeoff between the depth and spatial information.

(3) Figure 2c describes the coded aperture snapshot spectral imaging method (CASSI). The spectral data cube "$x + y + \lambda$" is dispersed by a prism and then modulated by a random coded mask, which results in a coded and sheared 3D data cube. The random coded mask is designed according to the Compressive Sensing (CS) principle, the complete "$x + y + \lambda$" data cube can be finely reconstructed based on the prior that spatial-spectral information is sparse in the wavelet domain.

To obtain high light throughput, high spatial resolution, and high spectral resolution at the same time, it is inevitable that spectral and spatial aliased images will be captured. Because spectral information and spatial information are coupled together, the acquisition of spectral information is transformed into solving an ill-posed optimization problem. The prior constraint is the basic technology for solving ill-posed optimization problems, and it is also the key to reconstructing space-spectrum coupled hyperspectral images. First, we establish a data fidelity term for the optimization target based on the physical model of the imaging process, and second, we establish a constraint term for the optimization target through the statistical

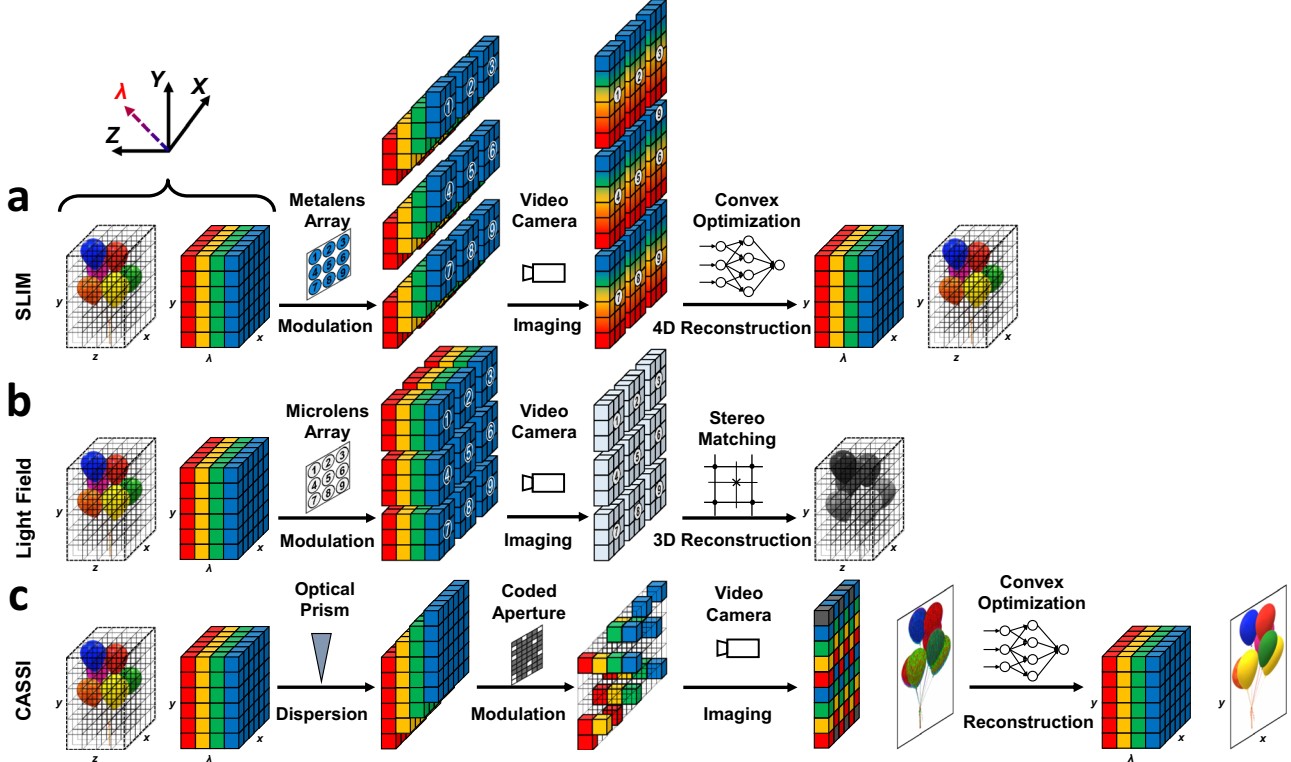

**Fig. 2 Data acquisition and sampling schemes of three computational imaging system.** The diagram of proposed spectral light field imaging system is shown in (**a**), the diagram of conventional light field imaging system is shown in (**b**), the snapshot spectral imaging system (CASSI) is shown in (**c**).

information of the real image. Because of the structure of the image and the consistency of the dispersion, we can always find a suitable optimal solution for the optimization goal.

**Imaging demonstration**. Figure 3 is a demonstration of the numerical simulation of our dispersion reconstruction algorithm to illustrate the feasibility of the algorithm. To this end, we randomly select a set of spectral data from the classic Columbia Multi-spectral Image Database[28], which contains a total of 31 spectral images from 400 to 700 nm, with 10 nm interval. We use the dispersion design of the proposed transversely dispersive metalens to generate a dispersion blurred image from these 31 spectral images, and then use our proposed spectral reconstruction algorithm to reconstruct 31 channels of spectral data from this dispersion blurred image. Here, we show five spectral images of the ground truth to compare with the reconstruction results to illustrate the effectiveness of our algorithm.

Figure 3a shows the original image of a piece of lemon. The simulated image related to the one captured by a transversely dispersive metalens is plotted in Fig. 3b. The transversely dispersive design of SLIM system focuses the different wavelengths onto different positions on the plane, forming a result like a motion blur image, with the useful spectral information hidden in the blur.

The proposed algorithm reconstructs the input dispersed gray image as a multi-spectral cube, to obtain clear texture information without dispersive blurring. The reconstructed spectral image can be estimated from an input dispersive blurred image[29], by minimizing the following convex optimization:

$$\operatorname*{argmin}_{S} ||\Phi S - D||_2^2 + \alpha_1 ||\nabla_{xy}S||_1 + \beta_1 ||\nabla_\lambda S||_1 \qquad (2)$$

The first term describes the data residual of our image formation model, and $||\cdot||_2$ is the L$_2$ norm, used to constrain the data fidelity.

$\Phi$ describe the image degradation from multi-spectral data to dispersed gray images. $S$ and $D$ are the spectral data and dispersed gray image, respectively. while the other terms are priors, and $\alpha_1$ and $\beta_1$ are the weights of the corresponding terms. $||\cdot||_1$ is the L$_1$ norm, where $\nabla_{xy}$ is a spatial gradient operator that denotes the difference between spectral data and the image plane, and $\nabla_\lambda$ is a spectral gradient operator that denotes the difference between spectral data at adjacent channels. The first prior is a traditional total variation term, ensuring sparsity of the spatial gradients and removal of the spatial artifacts. The second prior is a channel-wise total variation term, ensuring sparsity of the spectral gradients and preserving the spectral consistency.

The reconstructed gray image is shown in Fig. 3c. As shown in Fig. 3e, f, the reconstructed spectra are consistent with the ground truth at any part of the reconstructed image, such as the two points marked by the blue and red boxes in Fig. 3c. Therefore, the colorful image can be fully recovered (see Fig. 3d). It is important to see that both the color and spatial profile match the initial data very well. This consistency is further verified by comparing the ground-truth image (Fig. 3g) and reconstructed image (Fig. 3h) at different wavelengths.

With the transversely dispersive metalens and the spectrum reconstruction algorithm, the visual information of the scene including the spatial distribution of colorful objects can be recorded by a single snapshot of SLIM with a monochrome sensor. Figure 4a, b show a scene that consists of four letters "META" in different colors placed at different spatial positions. The measurement setup is shown in Supplementary Fig. 12. The raw data captured by a monochrome sensor is shown in Fig. 4c, which consist of 48 × 48 sub-images corresponding to different view angles. Each sub-image consists of 75 × 75 pixels and contains an inverted image of the letters (see the inset of Fig. 4c). Using the reconstruction algorithm, the spectral information of

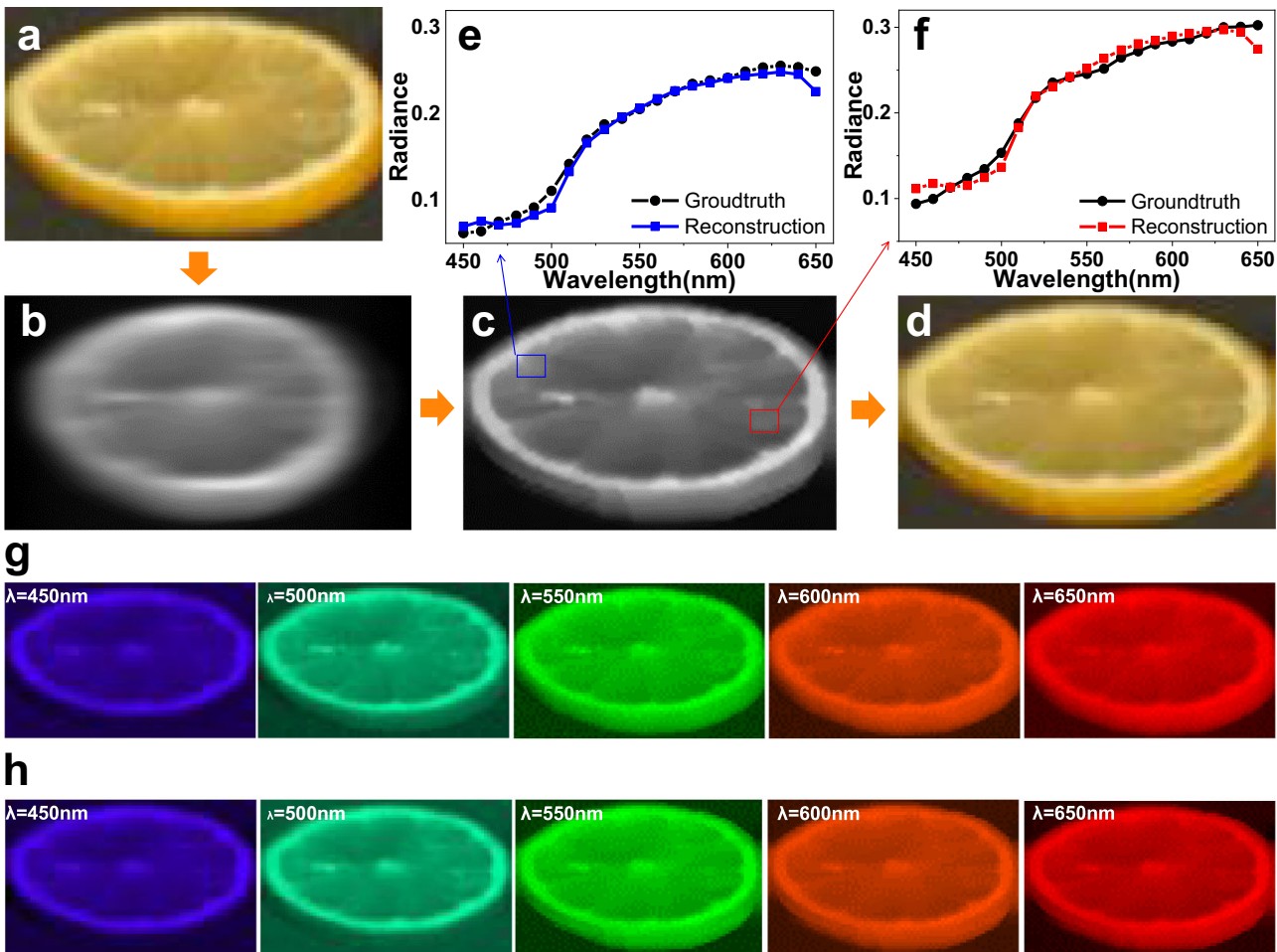

**Fig. 3 Numerical simulation results of spectral reconstruction algorithm for SLIM. a** The original RGB image of spectral data. **b** The simulated dispersion image using the forward model same as SLIM. **c** The reconstructed gray image. **d** The reconstructed color image synthesis from reconstructed spectral data. **e** The spectral plot of green position, the green line is reconstructed result, the black line is ground truth. **f** The spectral plot of blue position, the blue line is the reconstructed result, the black line is the ground truth. **g** The original single wavelength images are presented. **h** The reconstructed single wavelength images are presented.

each sub-image can be obtained with a spectral resolution of 8 nm (see Supplementary Note 4: Light-field-spectrum reconstruction algorithm for details). Following the conventional light-field imaging method, the 3D spatial information can be reconstructed using the sub-images[21]. The rendered all-in-focus image with a large depth-of-field is shown in Fig. 4d, in which all four letters are well imaged in terms of both the spatial positions and colors (spectral information). Figure 4e–h and Supplementary Movie 2 illustrate the reconstructed images at imaging depths of 69.8 cm, 57.3 cm, 45.1 cm, and 37.8 cm, which correspond to the actual positions of the four letters. Clear images of the four individual letters "M", "E", "T", and "A" with blurred backgrounds can be observed, presenting the ability to extract depth information with a metalens array. Furthermore, since SLIM can present the spectrum at any part of the images of the objects, it can successfully determine the exact colors of the objects. The spectra at arbitrary parts of the four letters are shown in Fig. 4i–l, which shows good agreement with the spectra measured by a commercial spectrometer, indicating good spectrum reconstruction. The 1951 United States Air Force resolution test chart has also been used to check the resolution of our SLIM system. Group 7, element 2 of the chart has been resolved (see Supplementary Fig. 11), giving a resolution (~2.9 µm) close to the Rayleigh criterion of the proposed metalens.

It should be noted that the spectral resolution of SLIM can be further improved by training the spectral super-resolution network on paired low-resolution spectral data and high-resolution spectral data (see Supplementary Note 5: Trained spectrum reconstruction algorithm for details), which are used for most optical measurements and material discriminations. Here, we demonstrate an imaging case beyond the ability of the naked eye and light-field imaging, which requires both high-resolution spectral information and depth information. The two kinds of materials, magenta chemical fabric cloth and water-color-painted paper, are shown in Fig. 5a, with their spectra plotted in the inset, which shows quite similar spectrum profiles in the visible regime. An "**I**" shaped chemical fabric cloth is placed a distance away from "**O**" shaped water-color-painted paper, as presented in Fig. 5b, c at different viewing angles. When a typical planar imaging camera is used, only a magenta "**Φ**" shaped image can be captured (see Fig. 5b), due to the lack of depth information and material properties from the high-resolution spectral information. Neither light-field imaging nor spectral imaging can totally reveal the difference between these two objects. Only SLIM imaging, which simultaneously obtains 4D information, can resolve this issue. As depicted in the inset of Fig. 5a, the spectra of the two materials have close peaks at 618 nm and 626 nm. After employing the trained spectrum reconstruction algorithm, a

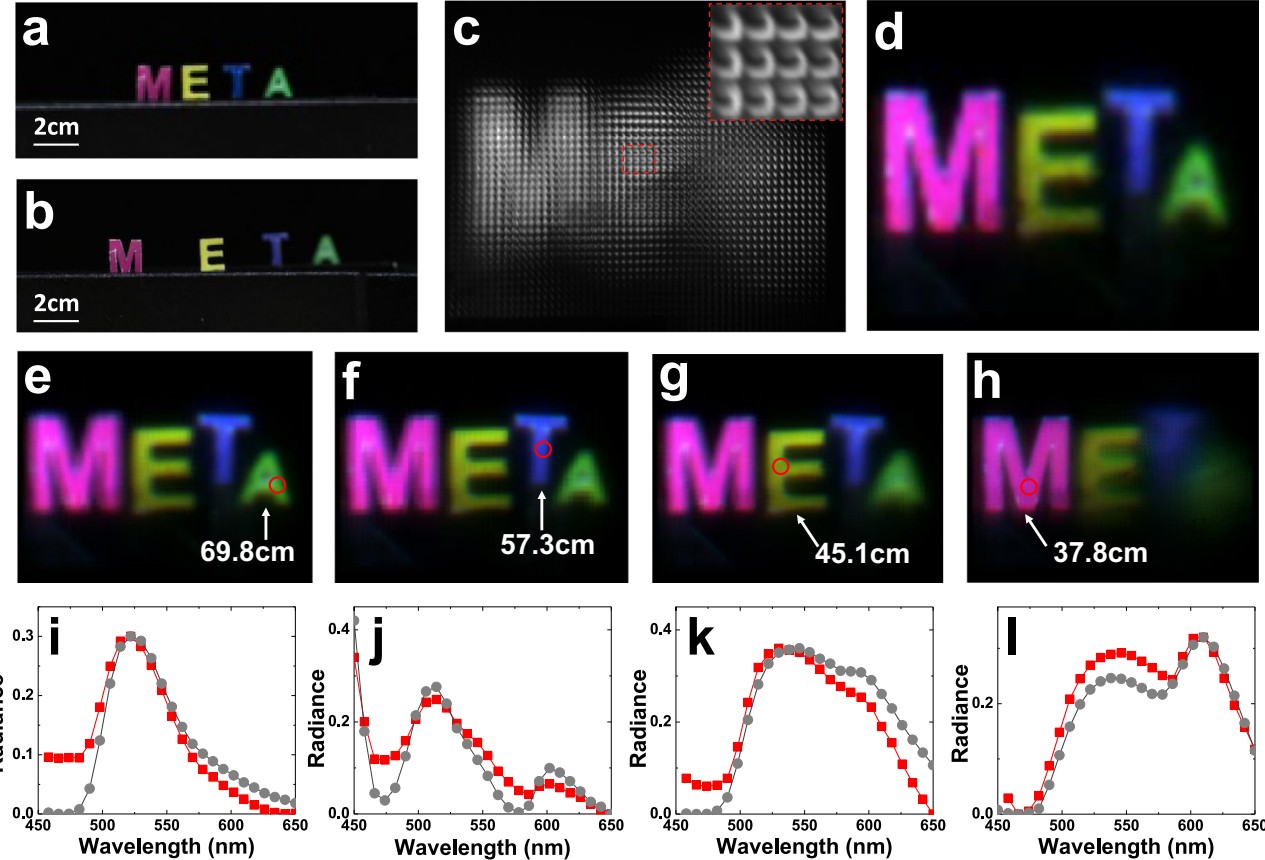

**Fig. 4 The spectral light-field imaging of metalens array. a, b** Different view of the scene that consists of four letters with different color at different spatial positions. **c** The raw data of the SLIM system. The inset is the Zoomed-in images of area inside the red box. **d** The all focused rendered color image. Rendered images with focusing depths of 69.8 cm (**e**), 57.3 cm (**f**), 45.1 cm (**g**) and 37.8 cm (**h**). **i-l** Spectra of four letters. The red dot lines are the spectra reconstructed from the metalens array, and the gray dot lines are measured by commercial spectrometer.

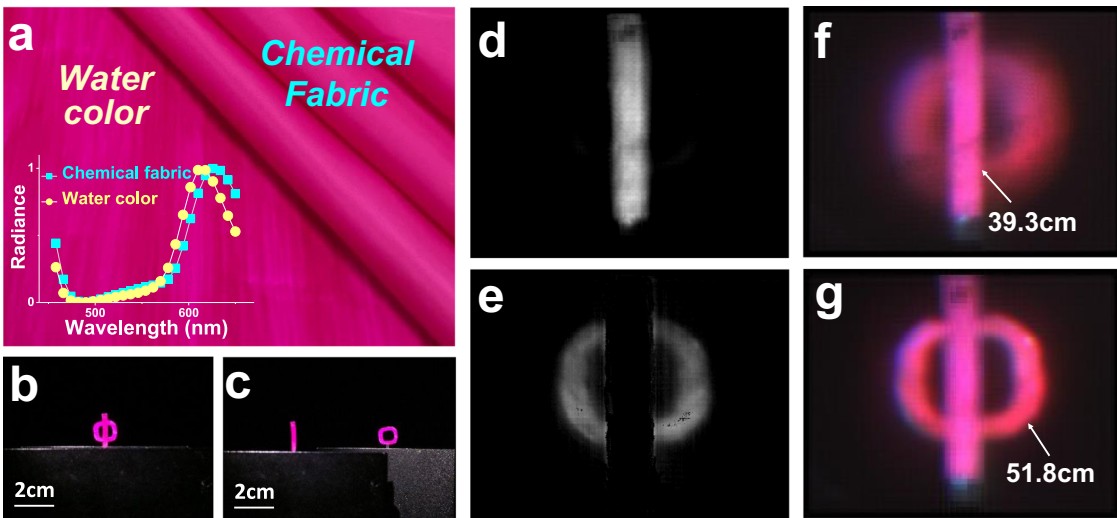

**Fig. 5 Material discrimination using SLIM. a** The watercolor painted paper and chemical fiber cloth. Inset is the spectrums of these two materials. **b, c** Different view of the scene that consists of "I"-shaped chemical fiber cloth and "O"-shaped watercolor painted paper placed away from each other. The material discrimination result for "I" (**d**) and "O" (**e**) respectively. **f** The rendered image at "I" plane. **g** The rendered image at "O" plane.

higher spectral resolution of up to 4 nm is achieved, at which the two spectral peaks can be well distinguished. After rendering the spectrum at any 3D spatial position in the scene, a simple process, $r(626) - r(618)$, can efficiently enlarge the difference between the

materials with spectrum peaks at 618 nm and 626 nm, where $r(\lambda)$ is the intensity at any spatial position with wavelength $\lambda$. Figure 5d shows the all-in-focus image after this process with normalization, from which one can clearly recognize the chemical

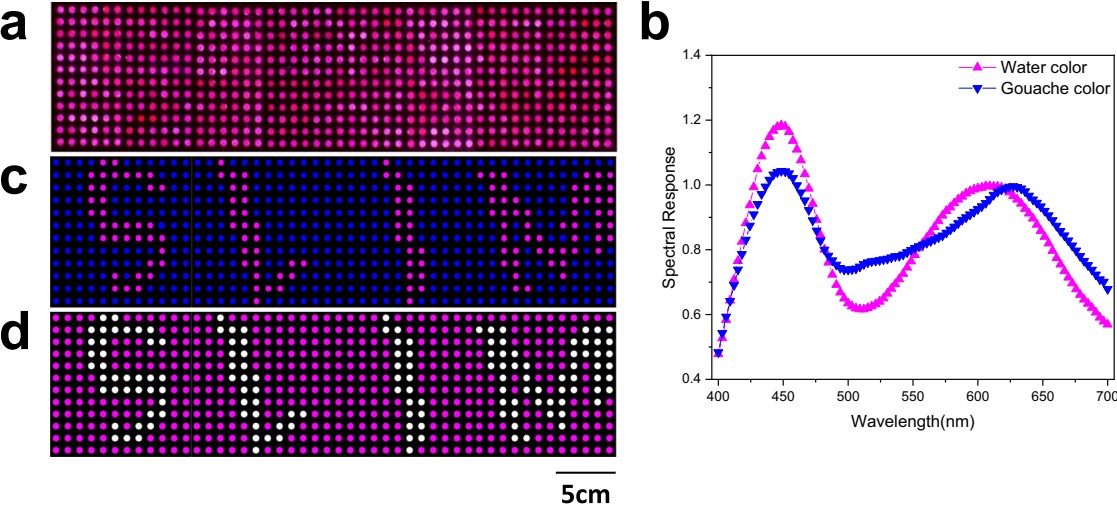

5cm

**Fig. 6 Material discrimination using SLIM. a** Images captured with conventional camera. **b** The spectral response for characters and background, respectively. **c** The rendered image with the SLIM system and reconstruction algorithm, the slight spectral difference can be magnified, making the hidden information visible. **d** Pattern design of (**a**).

fiber cloth from water-color-painted paper. In contrast, after inverse normalization with the larger intensity corresponding to a darker image, the paper painted by watercolor can be distinguished, (see Fig. 5e). Therefore, the high-resolution spectrum captured by SLIM can readily be used in material discrimination and camouflage cracking. At the same time, the light-field imaging enabled by the metalens array also presents the 3D spatial information of the two objects, as shown in Fig. 5f, g.

Until now, we have used the 4D information captured by the SLIM system to recognize objects with either different spatial positions or with almost the same color but different kinds of materials or textures, which surpasses the functionality of the naked eye, light-field imaging, and spectral imaging. For the same kind of material or texture, the spectra will be much more similar, and spectral recognition will be more challenging. Take the discrimination of two kinds of painting colors, e.g., gouache color and watercolor, as an example. We created a dot matrix on paper and painted the dots with these two colors to produce the information "SLIM", as shown in Fig. 6a. All dots have similar colors because of their very similar spectra, as shown in Fig. 6b. The unevenness that can be seen between the dots is due to the different thicknesses of the paint for each dot. It is quite difficult to extract hidden information via full-color imaging and even the above 4D imaging at a spectral resolution of 4 nm. A higher spectral resolution is required in this case. The proposed spatial-spectral-coupling sampling method transformed the tradeoff between the angular resolution, the spatial resolution, and the spectral resolution into a tradeoff between the angular resolution and the spatial resolution. For the FLF (focused light-field) scheme adopted by the SLIM, each micro-lens forms a relay system with the main lens. This configuration produces a flexible tradeoff in the sampling of spatial and angular dimensions and can more effectively sample the position information of the light field. Simply by changing the position of the micro-lens and the aperture of the main lens, the SLIM could flexibly switch the angular resolution, the spatial resolution, and the spectral resolution we need (see Supplementary Note 6: Tradeoff between in-plane spatial resolution, depth resolution, spectral resolution, angular resolution, and numerical aperture for detailed discuss). By adjusting the aperture of the main lens and the position of the metalens array, the object-image relationship can be changed, and

the spectral dispersion of the SLIM system can be flexibly tuned and even enlarged. A larger spectral dispersion leads to a higher spectral resolution. We also reduce the aperture size of the main lens to prevent image aliasing from the adjacent metalenses. Consequently, a spectral resolution of 3 nm can be obtained. Figure 6b illustrates the average spectra of the watercolor dots and gouache color dots from the samples, where slightly different variation trends of the spectra can be found from 525 to 625 nm. With a single snapshot of the SLIM system, the hidden information can be easily distinguished from the background. In Fig. 6c, the dots painted with watercolor and gouache color have been distinguished and marked in blue and pink, respectively. Here, the four characters of "SLIM" can be clearly seen, being very consistent with the original design (Fig. 6d).

## Discussion
The essence of the proposed SLIM is that the boundary constraints can be naturally shaped for each sub-aperture during the imaging through the metalens array (one device instead of many: transversely dispersive elements + code aperture/mask + microlens array), which harvests much more compactness and light throughput. In the proposed SLIM, the image is separated by each sub-aperture, which perform as another prior knowledge for the reconstruction. In Fig. 4, we are seeking simple demos to show potential application scenarios that bring public interest. In the more complicated case in Fig. 5, the capability of resolving the overlap of the spatial and spectral information has been demonstrated.

In summary, based on the transversely dispersive TiO$_2$ metalens, we have proposed and experimentally realized the first SLIM system that can simultaneously resolve 3D spatial information and additional spectral information. Objects with either slight spatial differences or spectral differences can be distinguished by rendering the sub-images via a spectrum super-resolution algorithm (see Supplementary Note 5: Trained spectrum super-resolution algorithm for detailed information). With this technique, a chameleon could be easily distinguished from the environment via the 4D information captured by the SLIM system. Note that SLIM is not limited to the visible transmission/reflection/emission spectrum. The same concept can be extended to infrared and Raman signals. Moreover, the compact SLIM is mainly realized within a thin metalens array, which is integrable with integrated optical systems

such as photonic chips or fibers. This 4D imaging capability of the metalens-array-based SLIM will revolutionize modern optical and bio-optical systems.

## Methods

**Sample fabrication**. Film deposition: The high-quality TiO$_2$ film is deposited onto a 13 nm indium tin oxide (ITO) coated glass substrate with electro-beam evaporator (Syskey A-75) with deposition rate of 0.8 Å/s and base vacuum pressure of $2 \times 10^{-7}$ torr. The optical parameters are measured using spectroscopic ellipsometry and shown in Supplementary Fig. 3. The real part of refractive index is above 2.1 through the whole visible wavelength while the loss is almost ignorable. Nanofabrication: After the deposition of TiO$_2$, 120 nm PMMA A2 is spin-coated onto the TiO$_2$ film and baked at 180 °C for 10 min. Then the PMMA film is exposed to electro-beam with an electron-beam lithography system (Raith E-line Plus). After developed in MIBK: IPA solution at 20 °C, the designed nanostructures are patterned in the PMMA film. Then the sample is transferred into an E-beam evaporator and 40 nm chromium (Cr) is deposited onto it. Finally, the pattern is transferred to Cr through lift off process using Remover PG.

By applying the Cr as the hard make, the TiO$_2$ film is etched with Cl$_2$ and Ar mixed gas in an inductively coupled plasma etcher (Oxford ICP100). The selectivity between Cr and TiO$_2$ is about 50:1 and the etching speed for TiO$_2$ is about 20 nm/s. The final TiO$_2$ metalens is achieved by removing the Cr mask in chromium etchant at room temperature for 2 min, as shown in Supplementary Fig. 4.

**Experimental setup**. After fabrication of the transverse dispersion metalens array, we carried out the characterization of the metalens light efficiency, the dispersion calibration of the metalens, the spatial resolution characterization, this is accomplished by using an optical relay system to image a pinhole or the 1951 USAF resolution test chart illuminated by fiber-coupled LEDs. We then conduct imaging experiments by replacing the pinhole with custom scenes. The fiber-coupled LEDs is used to illuminate the custom scenes that will be captured by our SLIM system. See Supplementary Note 3 for detailed description.

## Data availability

The data used to evaluate the spectral reconstruction algorithm is available with the paper. Full datasets are available from the authors upon reasonable request. Source data are provided with this paper.

## Code availability

The code used to evaluate the spectral reconstruction algorithm is available with the paper. Full codes are available from the authors upon reasonable request.

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

## Acknowledgements

The authors are grateful that this work was supported by the National Key R&D Program of China (2017YFA0303700), the National Natural Science Foundation of China (No. 62025108, 61627804, 11822406, 11774164, 11834007, 11774162, 11621091, 11974092). This work is also supported by the Fundamental Research Funds for the Central Universities No.020414380175, Shenzhen Fundamental research projects JCYJ20180507184613841, the University Grants Committee/Research Grants Council of the Hong Kong Special Administrative Region, China (Project No. AoE/P-502/20 and GRF Project: 15303521), the Shenzhen Science and Technology Innovation Commission Grant (No. SGDX2019081623281169), the Department of Science and Technology of Guangdong Province (2020B1515120073), and City University of Hong Kong (Project No. 9380131).

## Author contributions

X.H., Y.W. and S.W. contributed equally to this research. S.W., S.X., X.C. conceived the idea and supervise the research. X.H., Y.W, X.Z., Y.Z., L.L., F.Y. performed the optical experiment and designed the spectrum and light-field reconstruction algorithm. Y.W., S.X., J.H., performed fabrication of nanostructures and optical experimental measurements. X.H., Y.W., S.W., S.X., D.P.T., Z.W., S.Z. wrote the manuscript, and all the authors discussed the contents and prepared the manuscript.

## Competing interests

The authors declare no competing interests.
