## [Peer Review File · Nature Communications]

Reviewers' Comments:

Reviewer #1:

Remarks to the Author:

In this work, the authors demonstrate the single snapshot 4D imaging by using a newly proposed spectral light-field imaging (SLIM) technique. In this 4D imaging technique, the metalens array with strong optical dispersion was utilized to record both the spectral and spatial information simultaneously. The 4D imaging information can be acquired in the single snapshot mode with a monochrome CMOS sensor. They show that the visual indistinguishable objects and materials can be easily discriminated through the SLIM technique. The SLIM technique take the advantages of both strong dispersion of a single metalens and the conventional light field imaging system. I believe the proposed strategy may have important applications in fast and massive optical inspections for food, medicine, electronic devices and so on. I would support its publication after minor revisions.

Minor comments:

1. In Fig. 1B, there should be a typo, $\phi(b)$ - $\phi(b)$
2. It might be better to discuss the focusing efficiency of a single metalens;
3. In Figure 2G and 2F, the authors said "the original single wavelength images are presented", also in the main text, only a few words were used to discuss these two sub images. It is difficult to understand what does this sentence mean? In the main text, it is important to tell the reader how was the experiment was done to get these images, how do they reconstruct the images.

Reviewer #2:

Remarks to the Author:

The authors presented a snapshot spectral light-field imaging method using a dispersive metalens array. Although there are some novelties at the device level, they are not enough for journals like NC. Also, the necessity of using metalenses for spectral light-field imaging is not justified. For these reasons, I do NOT recommend its publication. The major critics are:

(1) To extract the spectroscopic information from the spectrally-blurred image behind each metalens, the authors used a deconvolution-like method described in a previous work (Baek, SH et al., ACM Trans. Graph. 36(6),1-2 (2017)), which was not appropriately cited in the main text. Like the use of deconvolution in removing the motion blur, this type of inverse problems is generally considered ill-posed and sensitive to the noise. Actually, it is for this reason that coded aperture snapshot spectral imager (CASSI) uses an encoded mask to make this inverse problem less ill-posed. In the current manuscript, there is no discussion on the robustness of their model to the noise and requirement on the scene sparsity. Although the simulation results demonstrated the reconstruction of a relatively complex scene (Fig.2), it was not showcased in experiments, where the authors intentionally chose sparse objects. For example, in Fig.S20(raw data of Fig.5), the scene is composed of point markers with equal spacing, which avoids significant overlap of the spatial and spectral information.

(2) At the device level, fabricating a dispersive metalens seems trivial, although may not in the sense of a controlled way. Metalenses naturally have strong chromatic aberrations because of the diffraction of nanoscale structures. The mainstream of the field is to correct for those aberrations and make metalenses achromatic. It is not clear whether their fabrication approach is necessary for the simple purpose of making the lens dispersive.

(3) At the system level, snapshot spectral light field imaging is not new. There are a few papers published on this topic (e.g. Opt. Express 26, 26495-26510 (2018)). Although the authors claimed that their system is compact and straightforward, it is at the expense of an increased computational cost. There is no discussion on this drawback. Also, the dispersive metalens array can be simply replaced by a conventional microlens array plus a transmission grating, a combination that can also be made ultracompact. Therefore, there is no such need to use complex metalenses for this purpose.

Reviewer #3:

Remarks to the Author:

This paper presents an extension of the concept of a metalens to include spectral dispersion. As a result, using a light-field imaging optical system, spectral information can be added to 3D light-field information to enable "4D" images (3 spatial dimensions plus one spectral dimension).

This is an important step forward for compact plenoptic imaging. The scope of the work and the significant advance in the field make the content of the work appropriate for Nature Communications. The work will likely be of broad interest to the optics and imaging communities.

While the content of the work is significant, the problem is that the paper cannot be published in its present form. Not to put too fine a point on it, the paper is simply extremely poorly written. Unfortunately, this is true on two levels. First, the English is simply not up to the required standard. The text requires major editing not only for style but also for comprehensibility. Second, and much more important, the presentation of the material is poorly organized and presented. The simple fact is that, with the present manuscript, the reader must spend an inordinate amount of time trying to figure out what the authors are trying to say. One has to go back and forth between different sections of the paper to try to piece together what the authors actually did and what the actual optical system is. There is no linear narrative that presents the system and its components in a way that is immediately understandable to anyone who is not already familiar with the results. Indeed, it feels like different sections of the paper were written by people involved with different aspects of the project, and then it was cut-and-pasted together.

For example, and most egregiously, the main text never actually shows a figure of how the whole system fits together. It APPEARS that the metalens is substituted for the microlens array in a conventional light-field camera configuration, so that each subaperture image also has spectral dispersion which can be pulled out using a reconstruction algorithm. However, the schematic in figure 1A shows an extended object imaged to three different spatial positions in the focal plane, which is quite different from the light-field camera microlens geometry. Furthermore, it is extremely difficult for the reader to deduce from the text or the figures what the actual geometry is. One of the contributing problems here is that none of the figures in the main or supplemental sections actually show how rays go from an object point to an image point and correspondingly to a pixel on the monochrome camera such that one can see how both spectral and angular (light-field) information are captured on the CMOS array. All of the figures show "schematic" pieces of rays, or show only a piece of a ray through a specific element, so it is impossible to trace an actual ray path. Examples include figures S10 and S12. The rays entering the lens don't come from object points, and they are not even bent by the main lens. If the rays converge as shown, then this is not a light-field configuration, since the lens array does not resolve the angular information in these rays.

I would go on to make more specific critiques and suggestions for particular sections of the discussion, but it is not worth it at this time. The paper first needs to be made readable, with a sequential/linear presentation of the concepts.

I will note that there are two important general considerations missing from the paper.

1. There is no discussion of the trade-offs between angular resolution, spatial resolution, and spectral resolution, given that there are only so many pixels on the monochrome detector array. Any light-field monochrome imaging system has a trade-off between spatial resolution and angular resolution (i.e. the number of pixels in each superpixel; for example, a 9x9 subarray on the detector reduces the number of pixels in the final image by $9 \times 9 = 81$). Now in this new system one adds spectral resolution. A simple-minded estimate would be that providing three colors (for example RGB channels) of spectral information would further reduce the image size by another factor of three. Nothing is for free, and these trade-offs need to be addressed.

2. There is no discussion of the uniqueness of the reconstruction. In other words, it appears (again, it is not presented clearly) that different object points will overlap on the sensor for different spectral channels, and that this is somehow deconvolved by an algorithm. The system clearly works for the examples presented in the paper, but it is not clear where the system might fail due to the spatial/spectral ambiguity. The lemon is not a hard problem; it is almost

monochrome yellow. The "META" object has different colors in it, but each simply-colored object is spatially well separated from the others. The data in figures 4 and 5 is more interesting (and deserve more discussion of how they work), but they are sufficiently simple that the results are quite plausible. The interesting question that I think should be addressed at some level is: when does the system fail? This is a linear detector, and there is an intrinsic ambiguity at a pixel level of whether a given intensity at that pixel is due to one spectral component coming from one object point, or a different spectral component coming from a shifted object point? Does the reconstruction algorithm produce a unique reconstruction always, or if not, under what conditions does it fail? (This is important to know both for practical applications and for determining how future research might overcome those limitations.)

I want to make it clear to the authors that I think they have done extremely interesting technical work. The problem is that the paper is organized and presented in such a way that it is extremely difficult for the reader to tell what's going on. After multiple readings of the paper, I still can't trace a ray through the system and tell how angular ray information is deconvolved from spectral information (unless perhaps the spectral components are very far separated). If they can fix that, and write a clear description of the work, then it will be acceptable and an important contribution.

Reviewer #1:

In this work, the authors demonstrate the single snapshot 4D imaging by using a newly proposed spectral light-field imaging (SLIM) technique. In this 4D imaging technique, the metalens array with strong optical dispersion was utilized to record both the spectral and spatial information simultaneously. The 4D imaging information can be acquired in the single snapshot mode with a monochrome CMOS sensor. They show that the visual indistinguishable objects and materials can be easily discriminated through the SLIM technique. The SLIM technique takes advantages of both the strong dispersion of a single metalens and the conventional lightfield imaging system. I believe the proposed strategy may have important applications in fast and massive optical inspections for food, medicine, electronic devices and so on. I would support its publication after minor revisions.

Our reply: We appreciate your positive comments on our work and have addressed all the concerns accordingly.

1. In Fig. 1B, there should be a typo, $\phi(b)$ - $\phi(b)$;

Our reply: Thanks for pointing out this mistake, the typo has been fixed in Fig. 1B.

2. It might be better to discuss the focusing efficiency of a single metalens;

Our reply: Thanks for this suggestion. The discussion of the focusing efficiencies at red (633nm), green (520nm), and blue (455nm) has been added into the Optical measurements section in the Supplementary information.

The following discussion has been added into Para 2, Page 3 in the Supplementary information:

Using laser light sources, the focus spot images and focus efficiencies at 633 nm, 520 nm, and 455 nm are measured, shown in Fig. S6. The efficiencies are 39%, 55%, and 72% (efficiencies are normalized to the incident light) for 633 nm, 520 nm, and 455 nm superior to the previous reports on TiO₂ metalens.

-
3. In Figure 2G and 2F, the authors said “the original single wavelength images are presented”, also in the main text, only a few words were used to discuss these two subimages. It is difficult to understand what does this sentence means? In the main text, it is important to tell the reader how was the experiment was done to get these images, how do they reconstruct the images.

Our reply: Thanks for the comments. The detailed description of the experiment and reconstruction algorithm has been refined and supplemented.

The revised manuscript interprets as follows in Para 1-3, Page 7:

Figure 2A shows the original image of a piece of lemon. The simulated image related to the one captured by transversely dispersive metalens is plotted in Fig. 2B. The transversely dispersive design of SLIM system focuses the different wavelengths onto different positions on the plane, forming a result similar to a motion blur image, with the useful spectral information hidden in the blur.

The proposed algorithm reconstructs the input dispersed gray image as a multi-spectral cube, to obtain clear texture information without dispersive blurring. The reconstructed spectral image can be estimated from an input dispersive blurred image (29), by minimizing the following convex optimization:

$$\underset{s}{\operatorname{argmin}} \|\Phi S - D\|_2^2 + \alpha_1 \|\nabla_{xy} S\|_1 + \beta_1 \|\nabla_\lambda S\|_1 \quad (1)$$

The first term describes the data residual of our image formation model, and $\|\cdot\|_2$ is the L-2 norm, used to constrain the data fidelity. Φ describe the image degradation from multi-spectral data to dispersed gray images. S and D are the spectral data and dispersed gray image, respectively. while the other terms are priors, α_1 and β_1 are the weights of the corresponding terms. $\|\cdot\|_1$ is the L-1 norm, where ∇_{xy} is a spatial gradient operator that denotes the difference between spectral data and the image plane, and ∇_λ is a spectral gradient operator that denotes the difference between spectral data at adjacent channels. The first prior is a traditional total variation term, ensuring sparsity of the spatial gradients and removal of the spatial artifacts. The second prior is a channel-wise total variation term, ensuring sparsity

of the spectral gradients and preserving the spectral consistency.

The reconstructed gray image is shown in Fig. 2C. As shown in Fig. 2E and Fig. 2F, the reconstructed spectra are consistent with the ground truth at any part of the reconstructed image, such as the two points marked by the blue and red boxes in Fig. 2C. Therefore, the color image can be fully recovered (see Fig. 2D). It is important to see that both the color and spatial profile match the initial data very well. This consistency is further verified by comparing the ground-truth image (Fig. 2G) and reconstructed image (Fig. 2H) at different wavelengths.

Reviewer #2:

The authors presented a snapshot spectral light-field imaging method using a dispersive metalens array. Although there are some novelties at the device level, they are not enough for journals like NC. Also, the necessity of using metalenses for spectral light-field imaging is not justified. For these reasons, I do NOT recommend its publication. The major critics are:

1. To extract the spectroscopic information from the spectrally-blurred image behind each metalens, the authors used a deconvolution-like method described in a previous work (Baek. SH et al., ACM Trans. Graph. 36(6),1-2 (2017)), which was not appropriately cited in the main text. Like the use of deconvolution in removing the motion blur, this type of inverse problems is generally considered ill-posed and sensitive to the noise. Actually, it is for this reason that coded aperture snapshot spectral imager (CASSI) uses an encoded mask to make this inverse problem less ill-posed. In the current manuscript, there is no discussion on the robustness of their model to the noise and requirement on the scene sparsity. Although the simulation results demonstrated the reconstruction of a relatively complex scene (Fig.2), it was not showcased in experiments, where the authors intentionally chose sparse objects. For example, in Fig.S20(raw data of Fig.5), the scene is composed of point markers with equal spacing, which avoids significant overlap of the spatial and spectral information.

Our reply: We first thank the reviewer for pointing out the inappropriate citation. We have moved it to the main text as Ref. [29].

More importantly, the discussions are replenished for the differences between our method, the previous deconvolution methods and CASSI. The key innovation of our method is as follows.

(1) Boundary constraint

In the latest work of our group [1], we found that, by applying the boundary constraints, without designing an elaborate coded-aperture, directly solving the dispersion-blurred image, it is still possible to accomplish as good reconstruction as coded-aperture scheme like CASSI. Different from the binary mask used in [1], *the*

essence of the proposed SLIM is that the boundary constraints can be naturally shaped for each sub-aperture during the imaging through the metalens array (one device instead of many: transversely dispersive elements + code aperture/mask + light field rendering + constraints helping the reconstruction), which harvests much more compactness and light throughput.

(2) Prior knowledge

The prior knowledge plays a key role in the reconstruction of the ill-posed problem. The frequently used prior knowledge is the sparsity of natural images, usually performed as the L-1 norm constraint as in most previous deconvolution methods and CASSI. In [1], one special prior is introduced as an element-wise weighting matrix that determines the level of confidence of the estimated gradients. *In the proposed SLIM, the image is separated by each sub-aperture , which performs as another prior knowledge for the reconstruction.*

(3) Robustness for more complex scene

As shown in Fig. R1, a more complex scene is used to verify the effectiveness and robustness of the proposed method incorporating new boundary conditions and prior. It has been added to the **Supplementary information** as Figure S22. We admit that gap factors still exist between the theory and real cases, same as CASSI reconstruction and practical CASSI cameras, because of fabrication, calibration, etc. As a result, in the real case in Fig. 3, we are seeking simple demos to show potential application scenarios that bring public interest. It needs to explain that the purpose of Fig. S20 and Fig. 5 is to show that SLIM can decrypt the hidden information with a finer spectral resolution. In that demo, equally spaced markers are only for the purpose of easy-reading. *In the more complicated case in Fig.4, the capability of resolving the overlap of the spatial and spectral information has been demonstrated, which is not a sparse case.*

Figure R1: Numerical simulation results of spectral reconstruction algorithm for SLIM. A. The original RGB image of spectral data. B. The simulated dispersion image using the forward model same as SLIM. C. The reconstructed gray image. D. The reconstructed color image synthesis from reconstructed spectral data. E. The spectral plot of green dot position in D, the green line is the reconstructed result, the black line is ground truth. F. The spectral plot of blue dot position in D, the blue line is the reconstructed result, the black line is the ground truth. G. The original single wavelength images are presented. H. The reconstructed single wavelength images are presented.

The following discussion has been added to the Discussion in Para 2, Page 12 in the main text:

The essence of the proposed SLIM is that the boundary constraints can be naturally shaped for each sub-aperture during the imaging through the metalens array (one device instead of many: transversely dispersive elements + code aperture/mask + light field rendering + constraints helping the reconstruction), which harvests much more compactness and light throughput. In the proposed SLIM, the image is separated by each sub-aperture, which performs as another prior knowledge for the reconstruction. the case in Fig. 3, we are seeking simple demos to show potential application scenarios that bring public interest. In the more complicated case in Fig.4, the capability of resolving the overlap of the spatial and spectral information

has been demonstrated, which cannot be considered as a sparse case.

2. At the device level, fabricating a dispersive metalens seems trivial, although may not in the sense of a controlled way. Metalenses naturally have strong chromatic aberrations because of the diffraction of nanoscale structures. The mainstream of the field is to correct for those aberrations and make metalenses achromatic. It is not clear whether their fabrication approach is necessary for the simple purpose of making the lens dispersive.

Our reply: It is true that metalenses naturally have strong chromatic aberrations with the focal spots moving along the propagational direction of light at different wavelengths. Nevertheless, such longitudinal chromatic aberrations are not helpful for the extraction of spectral information in a snapshot imaging system. The spectral information may be extracted from this longitudinal chromatic case by using the special algorithms, such as learning-based methods, however, the spectral resolution and accuracy cannot be guaranteed. To achieve higher spectral resolution and accuracy, the longitudinal chromatic aberrations needed to be compensated while the transverse chromatic aberration needed to be introduced and elaborately designed.

Our group has reported a series of works on broadband achromatic metalens to cancel this longitudinal chromatic aberration and achieved the various imaging systems based on broadband achromatic metalens (Ref. [18] and [19]). In spite of this, the achromatic phenomenon is not helpful for spectral imaging, because the differences between the wavelengths have been cancelled.

Since the transverse chromatic aberration, with images of different wavelengths spreading out on the imaging plane, brings more differences along with the varying wavelengths, which greatly benefits the extraction of the spectral information, in this work, we use the self-developed phase division principle (Ref. [18] and [19]) to design the metalenses with large transverse dispersion, which requires large phase compensation from the elements of the metalens. We employ the TiO₂ nano-pillars and nano-holes with high aspect ratios to work as the meta-elements to achieve this large phase compensation, which poses major challenges for conventional fabrication techniques such as Atomic Layer Deposition (ALD). In this sense, the proposed dry-etching approach is definitely necessary to fabrication these specially designed

metalens, and the fabrication technology and the sample's quality are both at the cutting-edge level.

3. At the system level, snapshot spectral light field imaging is not new. There are a few papers published on this topic (e.g. Opt. Express 26, 26495-26510 (2018)). Although the authors claimed that their system is compact and straightforward, it is at the expense of an increased computational cost. There is no discussion on this drawback. Also, the dispersive metalens array can be simply replaced by a conventional microlens array plus a transmission grating, a combination that can also be made ultracompact. Therefore, there is no such need to use complex metalenses for this purpose.

Our reply: We thank the reviewer for this important comment. As the reviewer mentioned, light-field spectral imaging has attracted widespread attention from the community, and there have been many excellent works, such as Opt. Express 26, 26495-26510 [OE 2018]. But these articles utilize additional optical components, for instance, the mentioned method [OE 2018] takes advantage of an additional Wollaston prism to achieve light field spectral information at the same time. The starting point of our SLIM method is that “*no additional optics should be required*”. All the imaging components lie in only an ultrathin planar optical element, which can not only minimize the calibration and alignment errors, but also improve the compactness of the entire system.

Considering the computational complexity, the SLIM system does not bring more than the previous works. Its computational complexity is similar to that of CASSI, which is based on mask-modulated compressed sensing algorithms. SLIM and CASSI both reconstruct the spectral data from the dispersive blurred image by solving an optimization target. The difference is that when establishing the data fidelity term, the CASSI needs to be multiplied by an additional mask modulation matrix, while the SLIM system directly establishes the relationship between the dispersion blur image and the spectral data through the dispersion transfer matrix. We have conducted all three types of calculations. To verify the computational cost, we perform experiments to reconstruct a 30-spectral channel, 512*512-spatial size data. The test platform uses i7-4790 CPU under Matlab® 2020b configuration. The relative tolerance of the

iteration termination is set to be $1e^{-6}$.

Computational Cost Test

Data: 30-spectral channel, 512*512-spatial size			
Algorithm	TWIST (CASSI)	SLIM	Baek. SH et al.
Reconstruction Time / min	39	38	45

The TWIST algorithm used by CASSI takes 39 minutes, the spectral reconstruction method of SLIM takes 38 minutes, while the method mentioned by the reviewer ([OE 2018]) takes 45 minutes. *From the practical experiment, we can see that the spectral reconstruction algorithm of SLIM does not raise additional computational complexity.* More tests can be released in the SI upon request.

Comparing to “conventional microlens array plus grating” scheme, metalens has much better performance. For the metalens, incoming light rays get to pass through metalens array and are directly received on the imaging sensor. On the other hand, the grating has multiple diffraction orders, leading to severe energy loss. To optimize the energy of the grating scheme, extra fabrication, such as blazed grating, is usually used, which raises a dilemma between the light throughput and the complexity of the system. In most instances, the “microlens array plus grating” scheme involves a relay lens to link the microlens array and grating, which sacrifices the compactness. If gratings must be used, it is best to integrate metalens and gratings, however, the fabricating difficulty of such a refraction-diffractive device is much higher than that of metalens.

As a matter of fact, we do have designed the refractive-diffractive grating structure to realize the SLIM imaging, before we submitted this manuscript at the beginning of 2021. We tried to fabricate such a sample by using the grey-level exposure technology (Heidelberg Maskless and Laser Lithography system). The typical measurement result of the refractive-diffractive grating sample is shown in Fig. R2a. The period of the grating is $45\ \mu\text{m} \times 45\ \mu\text{m}$, and the working band is from 450 nm to 750 nm with the transverse displacement of focal spots being $7\ \mu\text{m}$. Since the size of this sample is larger than the metalens used in our manuscript and the transverse chromatic aberration of the grating is smaller than that of the metalens, the fabrication of such grating should be much easier than that of metalens. However, the result is completely opposite to the expected outcome, see Fig. R2b. By using the state-of-the-art

technology, the structure details are still hard to be realized and good grating configuration can only be found at the center of the grating, which leads to the bad focusing and spectral splitting function, and evident loss from other order diffraction. In terms of fabrication difficulty, the planar metalens system is a much suitable and realistic choice to provide the large transverse dispersion.

Figure. R2: **a**. The height distribution of the grating; **b**. The height distribution along the white line in **a**, and the measure height distribution of the fabricated sample.

In summary, considering the above-mentioned *pros and cons*, the metalens array with large transverse dispersion is the best choice to realize the SLIM system at present.

Reviewer #3:

This paper presents an extension of the concept of a metalens to include spectral dispersion. As a result, using a light-field imaging optical system, spectral information can be added to 3D light-field information to enable “4D” images (3 spatial dimensions plus one spectral dimension).

This is an important step forward for compact plenoptic imaging. The scope of the work and the significant advance in the field make the content of the work appropriate for Nature Communications. The work will likely be of broad interest to the optics and imaging communities.

Our reply: We thank the reviewer for the positive comments on our work.

(1). While the content of the work is significant, the problem is that the paper cannot be published in its present form. Not to put too fine a point on it, the paper is simply extremely poorly written. Unfortunately, this is true on two levels. First, the English is simply not up to the required standard. The text requires major editing not only for style but also for comprehensibility.

Our reply: We thank the reviewer for this important suggestion. The manuscript is re-organized and extensively smoothed. In addition, the English is carefully checked and polished by the Springer Nature language editing service.

(2) Second, and much more important, the presentation of the material is poorly organized and presented. The simple fact is that, with the present manuscript, the reader must spend an inordinate amount of time trying to figure out what the authors are trying to say. One has to go back and forth between different sections of the paper to try to piece together what the authors actually did and what the actual optical system is. There is no linear narrative that presents the system and its components in a way that is immediately understandable to anyone who is not already familiar with the results. Indeed, it feels like different sections of the paper were written by people involved with different aspects of the project, and then it was cut and pasted together.

For example, and most egregiously, the main text never actually shows a figure of how the whole system fits together.

Our reply: We thank the reviewer for this careful review. Systematic decomposition animation has been added to the Supplementary materials (4D imaging system.mp4), which shows a clear composition of the SLIM system. The organization and the written have also been modified thoroughly.

(3). It APPEARS that the metalens is substituted for the microlens array in a conventional light-field camera configuration, so that each sub-aperture image also has spectral dispersion which can be pulled out using a reconstruction algorithm. However, the schematic in figure 1A shows an extended object imaged to three different spatial positions in the focal plane, which is quite different from the light-field camera microlens geometry.

Our reply: We thank the reviewer for this critical comment. In the proposed SLIM system, it is the transversely dispersive metalenses that enable the extraction of spectral information. The purpose of Figure 1A is to demonstrate the transversely dispersive focusing of single metalenses, not for light-field camera. By using the metalens array configuration, the images from different viewing angles can be captured, which realizes the light-field imaging. Following the comment, we have clarified the description in the caption of Figure 1A in the manuscript.

(4). Furthermore, it is extremely difficult for the reader to deduce from the text or the figures what the actual geometry is. One of the contributing problems here is that none of the figures in the main or supplemental sections actually show how rays go from an object point to an image point and correspondingly to a pixel on the monochrome camera such that one can see how both spectral and angular (light-field) information are captured on the cmos array. All of the figures show “schematic” pieces of rays, or show only a piece of a ray through a specific element, so it is impossible to trace an actual ray path. Examples include figures S10 and S12. The rays entering the lens do not come from object points, and they are not even bent by the main lens.

If the rays converge as shown, then this is not a light-field configuration, since the lens array does not resolve the angular information in these rays.

Our reply: Thanks for raising this point. In order to address this concern, the **ray path tracing of the SLIM** is obtained by using Zemax. We simulate the light-field imaging at a certain wavelength. The ray paths in the SLIM system with different incident angles of 1° , 2° , and 3° , are plotted in Fig. R3-1, which resolves the angular information.

Figure R3-1: The actual ray tracing path of SLIM system. **a.** separate ray tracing path for 450 nm, 550 nm, 650 nm. **b.** separate dot map at imaging plane.

Figure R3(b) shows the separate dot maps at the imaging plane, where the dots have an obvious displacement between different wavelengths indicating the transverse dispersion of metalenses. Since Zemax cannot import different phase distributions at different wavelengths, we cannot obtain the spectral light-field simulation at once. Therefore, we don't put the spectral ray tracing simulation in our manuscript, but readers can refer to the video (4D imaging system.mp4) in Supplementary information for a better understanding of the complete light path diagram of the SLIM system.

For light-field cameras, they are divided into two kinds, one is that each micro-lens does not form a separate image, named the unfocused light-field (ULF) camera, which was first proposed in [3] and improved in [4]; the other one is that each micro-lens forms a separate image, named as the focused light-field (FLF) camera, proposed in [2]. Fig. R3-2 illustrates the optical setup comparison between the ULF camera and the FLF camera.

To address this concern, a comparison is provided between the ULF and the FLF below:

- (1) lateral resolution. The ULF camera has a wider depth range but lowers lateral resolution than those of a FLF camera when the working distances of the two cameras are close.
- (2) depth resolution. Additionally, the ULF camera outperforms the FLF camera in depth resolution regarding the maximum number of resolvable steps in their depth ranges [5]. However, the reconstruction accuracy of an ULF camera is worse than that of a FLF camera.
- (3) spectral resolution. The FLF camera can adjust the trade-off between spectral resolution and lateral resolution by controlling the distance of the microlens to control the transversely dispersive distance.

In this way, the camera captures dense positional information, rather than capturing dense directional information, resulting in spatial resolution that is significantly higher than the number of micro lenses used. Based on this fundamental comparison and the demand for better lateral resolution for spectral imaging, we choose focused light-field (FLF) camera setup for the SLIM system.

Figure R3-2: comparison between the ULF setup and the FLF setup.

(5). I would go on to make more specific critiques and suggestions for particular sections of the discussion, but it is not worth it at this time. The paper first needs to be made readable, with a sequential/linear presentation of the concepts. I will note that there are two important general considerations missing from the paper.

There is no discussion of the trade-offs between angular resolution, spatial resolution, and spectral resolution, given that there are only so many pixels on the monochrome detector array. Any light-field monochrome imaging system has a trade-off between spatial resolution and angular resolution (i.e. the number of pixels in each superpixel; for example, a 9×9 subarray on the detector reduces the number of pixels in the final image by $9 \times 9 = 81$). Now in this new system one adds spectral resolution. A simple-minded estimate would be that providing three colors (for example RGB channels) of spectral information would further reduce the image size by another factor of three. Nothing is for free, and these trade-offs need to be addressed.

Our reply: We thank the reviewer for this important comment. We build the SLIM system using the principle of FLF camera, proposed by Lumsdaine [2]. The FLF camera Like the traditional light-field camera, uses an array of microlenses internal to the camera to capture radiance. Nevertheless, as the name implies, the FLF camera

uses the microlens array as an array of micro cameras, each of which captures a focused micro image. In this way, the new camera captures dense positional information, rather than capturing dense directional information, resulting in spatial resolution that is significantly higher than the number of microlenses used. For FLF camera, the spatial angular resolution depends on the depth of focus, different depth of focus, different resolution, which is always fixed for the ULF camera.

For the acquisition of spectral information, unlike the traditional scanning or filtering strategy, The SLIM system uses a computational algorithm to reconstruct the spectral signal from the dispersed blurred image. For example, If the size of a non-dispersive image is 512×512 , then the size of a dispersive image that can reconstruct 30 channels of spectral information will be $512 \times (512 + 30)$, which is only 30 columns more than the non-dispersive image. In short, only the last columns take 30 more pixels, the dispersion information of other pixels is coupled and superimposed on neighbor pixels.

We have revised the manuscript and insert this discussion as the reviewer suggested in Para 2, Page 11.

The proposed spatial-spectral-coupling sampling method transformed the trade-off between the angular resolution, the spatial resolution, and the spectral resolution into a trade-off between the angular resolution and the spatial resolution. For the FLF (focus light-field) scheme adopted by the SLIM, each micro-lens forms a relay system with the main lens. This configuration produces a flexible trade-off in the sampling of spatial and angular dimensions, and can more effectively sample the position information of the light field. Simply by changing the position of the micro-lens and the aperture of the main lens, the SLIM could flexibly switch the angular resolution, the spatial resolution, and the spectral resolution that are required in the practice.

(6). There is no discussion of the uniqueness of the reconstruction. In other words, it appears (again, it is not presented clearly) that different object points will overlap on the sensor for different spectral channels, and that this is somehow deconvolved by an algorithm. The system clearly works for the examples presented in the paper, but it is

not clear where the system might fail due to the spatial/spectral ambiguity. The lemon is not a hard problem; it is almost monochrome yellow. The “META” object has different colors in it, but each simply-colored object is spatially well separated from the others. The data in figures 4 and 5 is more interesting (and deserve more discussion of how they work), but they are sufficiently simple that the results are quite plausible. The interesting question that I think should be addressed at some level is: when does the system fail? This is a linear detector, and there is an intrinsic ambiguity at a pixel level of whether a given intensity at that pixel is due to one spectral component coming from one object point, or a different spectral component coming from a shifted object point? Does the reconstruction algorithm produce a unique reconstruction always, or if not, under what conditions does it fail? (This is important to know both for practical applications and for determining how future research might overcome those limitations.)

Our reply: We thank the reviewer for the suggestion to make the reconstruction process clear. To obtain high-resolution spatial information and spectral information at the same time, it is necessary to take a dispersion-blurred image, i.e., the spectral information and spatial information are coupled together. Then, as the reviewer expects, a deconvolution algorithm must be required. Priori constraint is the common strategy for solving such ill-posed deconvolution problems, and it is also the key to reconstructing space-spectrum coupled hyperspectral images. Firstly, we established a *data fidelity term* for the optimization target function based on the physical model of the imaging process. Secondly, we established a constraint term through the statistical information of the natural images and the prior. Because of the structure of the image and the consistency of the spectral dispersion, a suitable optimal solution is tended to be converged by iterate the optimization target function, however, failure cases truly exist mostly for two reasons:

(1) Calibration error: the deconvolution algorithm relies highly on the accuracy of the calibration of the dispersed wavelength position on the image sensor. Calibration error leads to a wrong mapping between the captured space-spectrum coupled image and the expected true data. Miscalibration of transverse dispersion leads to spectral reconstruction failure, see Fig. R3-3.

Figure R3-3. Miscalibration of transverse dispersion leads to spectral reconstruction failure. a. successful reconstruction. b. 3 pixels off correct dispersion calibration. c. 6 pixels off correct dispersion calibration. d. 9 pixels off correct dispersion calibration.

(2) Parameter choice: the weight parameters play an importance role during the optimization process. Currently, the parameters are set empirically, wrong choice causes the unbalance between the *data fidelity term* and the *prior*, resulting in failure reconstruction. In the near future, a robust parameter choice could be achieved by learning a large spectral database.

We have revised the manuscript on the above issues as the reviewer suggested in Para 2, Page 7.

In order to obtain high light throughput, high spatial resolution, and high spectral resolution at the same time, it is inevitable that spectral and spatial aliased images will be captured. Because spectral information and spatial information are coupled together, the acquisition of spectral information is transformed into solving an ill-posed optimization problem. The prior constraint is the basic technology for solving ill-posed optimization problems, and it is also the key to reconstructing space-spectrum coupled hyperspectral images. First, we establish a data fidelity term for the optimization target based on the physical model of the imaging process, and second, we establish a constraint term for the optimization target through the statistical information of the real image. Because of the structure of the image and the consistency of the dispersion, a suitable optimal solution can be achieved for the optimization function.

Reference

- [1] Zhao, Y., et al. "Spectral Reconstruction From Dispersive Blur: A Novel Light Efficient Spectral Imager." 2019 IEEE/CVF Conference on Computer Vision and Pattern Recognition (CVPR) IEEE, 2019.
- [2] Lumsdaine, Andrew, and Todor Georgiev. "The focused plenoptic camera." 2009

IEEE International Conference on Computational Photography (ICCP). IEEE, 2009.

[3] E. H. Adelson and J. Y. A. Wang, "Single lens stereo with a plenoptic camera," in IEEE Transactions on Pattern Analysis and Machine Intelligence (IEEE, 1992), pp. 99–106.

[4] R. Ng, "Digital light field photography," Ph.D. thesis (Stanford University, 2006).

[5] Shuaishuai Zhu, Andy Lai, Katherine Eaton, Peng Jin, and Liang Gao, "On the fundamental comparison between unfocused and focused light field cameras," Appl. Opt. 57, A1-A11 (2018)

Reviewers' Comments:

Reviewer #2:

Remarks to the Author:

My comments have been satisfactorily addressed. So I recommend its publication as is.

Reviewer #3:

Remarks to the Author:

The authors have added some detail in answer to the referee questions.

However, even with the revisions, the paper is still **extremely** hard to read and follow. It is just not organized in a way that makes it possible to understand the optical system, the ray paths, the mechanism of image formation, and the reconstruction in a straightforward way. Perhaps more importantly, the authors still do not address the tradeoffs caused by the introduction of spectral information to the conventional gray-scale light-field information. There are only so many pixels in the CMOS sensor; in a conventional monochrome LF camera there is a tradeoff between spatial and angular resolution that is well known. Now spectral information is added, with the number of sensor pixels fixed. Are the authors claiming that one gets 4-nm spectral resolution for free, at no cost to spatial and angular resolution? They present examples that seem to work, but they are quite sparse in at least one dimension, so it is not clear what the resolution tradeoffs are at all.

Response to Reviewer 3:

The authors have added some detail in answer to the referee questions. However, even with the revisions, the paper is still **extremely** hard to read and follow. It is just not organized in a way that makes it possible to understand the optical system, the ray paths, the mechanism of image formation, and the reconstruction in a straightforward way.

Our reply:

We thank the reviewer for the comment. The manuscript is re-organized and extensively smoothed. Two elaborately designed illustrations are added to provide a better understanding of the proposed SLIM system, see Fig. R1 and Fig. R2. We sincerely hope this can make it easier to get a big picture about the optical system, the mechanism of image formation and the reconstruction procedure of SLIM.

Figure R1: Optical paths of different light-field and spectral imaging schemes.

Figure R2: Data acquisition, sampling principles and reconstruction schemes.

For better understanding of our SLIM system, we compare it with the conventional light field imaging¹ and snapshot spectral imaging system². The optical paths of these three systems are shown in Fig. R1. The imaging mechanisms and reconstruction schemes of three systems are presented in Fig. R2.

(1) As for the SLIM system, the 4D ($x + y + z + \lambda$) data cube is modulated by metalens array, and decoupled into multi-view “ $x + y + \lambda$ ” information. Then, due to the high dispersion of the proposed metalens, blurred image (caused by the large transverse dispersion of metalens) is formed behind each metalens and captured by the camera. Finally, utilizing the proposed light field spectrum reconstruction algorithm (an inherent convex optimization method), both “ $x + y + \lambda$ ” and “ $x + y + z$ ” imaging result of the captured scene can be recovered, shown in upper part of Fig. R2.

(2) The middle part of Fig. R2 describes the light field imaging, which lacks the capability of encoding spectral information, layered depth images (LDI) can be obtained with the tradeoff between the depth and in-plane spatial information.

(3) The bottom part of Fig. R2 describes the coded aperture snapshot spectral

¹ Ng, R. Fourier slice photography. ACM Trans. Graphics 24(3), 735-744 (2005).

² Wagadarikar, A. et al., Single disperser design for coded aperture snapshot spectral imaging. Appl. Opt. 47(10), B44-B51 (2008).

imaging method (CASSI). The spectral data cube ($x + y + \lambda$) is dispersed by a prism and then modulated by a random coded mask, which results in a coded and sheared 3D data cube. The random coded mask is designed according to the Compressive Sensing (CS) principle, the complete “ $x + y + \lambda$ ” data cube can be finely reconstructed based on the prior that spatial-spectral information is sparse in the wavelet domain.

We have revised the manuscript and added Figure R2 to the main text of manuscript as Figure 2. Description of data acquisition, sampling principles and reconstruction schemes has been added to the main article, at Page 7 marked by red color.

Figure R3: Detailed illustration of the SLIM image formation.

Finally, the entire process of imaging of an object, a letter “A” with multiple wavelengths of light, through the SLIM system is shown in Fig. R3. The scene is first imaged through the main lens (part-1), then relayed by the metalens array for secondary imaging (part-2). Each metalens captures a part of the scene due to the difference in position (the images captured by adjacent metalens also have overlapping parts to calculate the depth information, part-3). Since the proposed metalens is designed to have transverse dispersion, each metalens forms a dispersive blurred image. The images with different wavelengths have small displacements from each other in the lateral direction, so they are superimposed leading to the blurred

image on the camera. Taking advantage of the additional information brought by this blurry, we could reconstruct the spectral information from the images of each metalens. After the spectral information reconstruction is completed, the clear spectrum images of each metalens is achieved. Then, using the obtained spectral images, the depth information can be obtained by calculating the parallax between adjacent metalens. At this time, both spectral and depth reconstruction can be achieved.

Perhaps more importantly, the authors still do not address the tradeoffs caused by the introduction of spectral information to the conventional gray-scale light-field information. There are only so many pixels in the CMOS sensor; in a conventional monochrome LF camera there is a tradeoff between spatial and angular resolution that is well known. Now spectral information is added, with the number of sensor pixels fixed. Are the authors claiming that one gets 4-nm spectral resolution for free, at no cost to spatial and angular resolution? They present examples that seem to work, but they are quite sparse in at least one dimension, so it is not clear what the resolution tradeoffs are at all.

Our reply:

We thank the reviewer for pointing out this important issue. Careful analysis and derivations are conducted to demonstrate the tradeoffs between the in-plane spatial, spectral and depth resolution. Before analyzing the tradeoffs between in-plane spatial, spectral, and depth resolution, we first define the resolution of these three variables, to make the discussion simple and clear.

- (1) The in-plane spatial resolution is defined with pixel counts. For example, an image with $R_{spatial}$ pixels height and $R_{spatial}$ pixels width, is defined with an in-plane spatial resolution of $R_{spatial}$.
- (2) The spectral resolution is defined with spectral channel numbers, $R_{spectral}$. We assume that the dispersion is approximately uniform, which is consistent with our

design. Therefore, a 50-channel spectral image from SLIM system (ranging from 450-650 nm) yields an actual spectral resolution of 4 nm.

- (3) The depth resolution is defined with depth layers, R_{depth} . For example, a 10-layer slices of scenes ranging from position 20 cm to 30 cm yields an actual depth resolution of 1 cm.

1. Tradeoff between the in-plane spatial resolution and the spectral resolution

The spectral information is reconstructed from the dispersion blurred image, and the image formation process is shown in Fig. R4. The light of different wavelengths from a single pixel is dispersed by metalens and transmitted to different positions on the imaging plane. The spectral information of different wavelengths of different pixels is superimposed (integrated) and captured by the sensor to form a dispersion blurred image. Therefore, when the size of the single-wavelength image is $R_{spatial} * R_{spatial}$, the channel size is $R_{spectral}$, the size of the dispersion blurred image captured by the camera will be $R_{spatial} * (R_{spatial} + R_{spectral})$, shown in Fig. R4. While the size of the reconstructed spectral data cube will be $R_{spatial} * R_{spatial} * R_{spectral}$.

Figure R4: Dispersion image formation with transversely dispersive metalens.

Because the proposed metalens array is arranged in a tight form, the in-plane spatial resolution + spectral resolution of the image under a single metalens should be

not greater than the lens diameter. The diameter of single metalens is denoted by D_{lens} , and the diameter of single metalens at imaging plane is proportional to D_{lens} . If the pixel size is denoted by P , the space bandwidth product of single metalens can be expressed as $\frac{D_{lens}}{P}$, indicating the total information. Therefore, the relationship between in-plane spatial resolution and spectral resolution is:

$$R_{spectral} + R_{spatial} = \xi \frac{D_{lens}}{P} \quad (1)$$

where ξ is related with the size of the image of the metalens. *This relationship shows that the summation of the in-plane spatial and spectral resolution is limited by the size of image captured by each metalens.*

2. Tradeoff between depth resolution and spectral resolution

According to the imaging principle of the focused light field (FLF) camera, the depth resolution of a SLIM camera is related to the optional patch size of a single metalens. The final image is rendered by tiling the selected squares together. Choosing one pitch or another puts different world planes “in focus.” In other words, patches match each other perfectly only for one image plane behind the main lens. By the lens equation, this corresponds to a given depth in the real world. Thus, a different patch size would correspond to a different depth. The equivalent of refocusing is accomplished in the FLF camera data through the choice of the patch size. Hence, the maximum depth resolution is half of the space bandwidth product $\frac{D_{lens}}{P}$ for the symmetry of the lens:

$$R_{depth} = \frac{\xi}{2} \frac{D_{lens}}{P} \quad (2)$$

In fact, the relationship between in-plane spatial resolution, spectral resolution and depth resolution is obtained by combining Eq. (1) and (2) as follow:

$$\eta(R_{spectral} + R_{spatial}) + 2(1 - \eta)R_{depth} = \xi \frac{D_{lens}}{P} \quad (3)$$

Here, η is an arbitrary positive coefficient smaller than 1, which evidently shows the tradeoff between in-plane spatial resolution, spectral resolution, and depth

resolution. Moreover, with a fixed number of pixels of the camera, Equation (3) shows the limit of the summation of these different resolutions.

Description of Tradeoff between in-plane spatial resolution, spectral resolution, and depth resolution has been added to the main text, at Page 12 marked by red color.

3. Tradeoff between spectral resolution and numerical aperture

In addition to the above-mentioned tradeoff relationship, by adjusting the position of metalens array, the object-image relationship is altered, and the image distance is extended. Therefore, we could obtain a larger dispersion width, thereby improving the spectral resolution of the imaging system, shown in Fig. R5.

Figure R5: Extended spectral resolution by changing the imaging distance. Where f is the focal length and b is the imaging distance.

Here, the distance between the focal points of 450 nm and 650 nm in the focal plane is w_0 , the dispersed spectrum width w_i on imaging plane would be:

$$w_i = \frac{b}{f} w_0 \quad (4)$$

In SLIM system, the spectral resolution is determined by the dispersed spectrum width on the sensor plane, which means the spectrum width varies over the image planes. The spectral resolution at imaging plane is:

$$R_{\text{spectral}} = \frac{w_i}{P} = \frac{b}{f} \frac{w_0}{P} \quad (5)$$

Due to the tight arrangement of metalens arrays, simply increasing the image distance in exchange for higher spectral resolution will cause imaging aliasing between adjacent metalens, shown in Fig. R6.

Figure R6: Image formation of SLIM. The object light forming the aliased image is indicated by the dotted line and the aliased areas are represented by transparent images. Object light causing aliasing is blocked by the diaphragm.

Here, the imaging distance is b . To prevent the aliasing between adjacent metalens, the size of the image should not exceed the diameter of the lens, which requires the cone angle of the object ray bundles to meet the following conditions. The numerical aperture NA of an objective lens is defined by:

$$NA = n \sin[\theta], \quad (6)$$

where n is the refraction index of the surrounding medium. Here, it is air, $n=1$. From Eq. (4) and Eq. (5), we derive the spectral resolution $R_{spectral}$ for SLIM with respect to the numerical aperture is:

$$R_{spectral} = \frac{D_{lens} w_0 n}{2f P NA} \quad (7)$$

By adjusting the numerical aperture of the imaging system and the position of the metalens array, in order to furthest extend the spectral resolution, the depth and in-plane spatial information should be compressed, as shown in the supplementary material Fig. S23. This evidently shows the tradeoff between the spectral resolution and spatial resolution.

Figure S23: Enlarged dispersion width. A. Raw data of Figure 5; B. Position of 450 nm, 550 nm and 650 nm spot are marked in blue, green, and red, respectively.

Increasing the resolution of any dimension will cause the loss of the resolution of the other two dimensions. Among them, due to the strong coupling of spectral information and in-plane spatial information, when the in-plane spatial resolution is improved, a subtle spectral resolution loss will occur, and vice versa. But for the depth resolution, the situation will be more complicated. Recording angular information requires a huge number of pixels. Even if we use the FLF setting, coupling the spatial information and angle information together. Hence, when trying to increase the angular resolution, it will cause a large reduction of in-plane spatial and spectral resolution.

4. Tradeoff between angular resolution and spectral resolution

The angular samples for the focused light field camera for a given spatial point are obtained by different metalenses. Hence, the angular resolution $R_{angular}$ for the focused light field camera is determined by the optical geometry \mathbf{a} and \mathbf{b} , the distance from spatial point to the metalens and the distances from metalens to imaging sensor.

Figure R7: intermediate image space of focused light field camera

Figure R7 shows the intermediate image space of a focused light field camera. The orange, green and blue lines are the chief rays associated with metalens L1, L2, and L3, respectively. Image points between plane v_i and v_{i-1} can be imaged by i microlens, where plane v_i and v_{i-1} have a distance b^3 . In this case, the $R_{angular}$ of the light field imaging system for point A is i . Figure R7 corresponds to the case $i = 2$. The $R_{angular}$ of focused light field camera could be defined as follow:

$$R_{angular} = \frac{a}{b} \quad (8)$$

Here, a represent the distance from spatial point to the metalens, b represent the distances from metalens to imaging sensor. The relationship between a and b could also be described with Gaussian lens formula, shown as follow:

³ Zhu, Shuaishuai, et al. "On the fundamental comparison between unfocused and focused light field cameras." Applied optics 57.1 (2018): A1-A11.

$$\frac{1}{a} + \frac{1}{b} = \frac{1}{f_{metalens}} \quad (9)$$

Therefore, combining **Eq. 8** and **Eq. 9**, the $R_{angular}$ could be expressed as:

$$R_{angular} = \frac{f}{b - f} \quad (10)$$

From the analysis in the previous section, we show that the spectral resolution $R_{spectral}$ of the SLIM system is closely related to the imaging distance b . Considering **Eq. 5**, the relationship between $R_{spectral}$ and $R_{angular}$ could be derived as:

$$R_{angular} = \frac{1}{\frac{P R_{spectral}}{w_0} - 1} \quad (11)$$

To prevent image aliasing between adjacent lenses, when adjusting the imaging distance, the NA of SLIM system must also be adjusted to match the imaging distance. From **Eq. 7** and **Eq. 11**, the relationship between $R_{angular}$ and NA can be derived as:

$$R_{angular} = \frac{1}{\frac{n D_{lens}}{2f NA} - 1} \quad (12)$$

Equation 11 and 12 show the tradeoff between $R_{angular}$, NA , and $R_{spectral}$ that the larger $R_{spectral}$ or the smaller NA will lead to smaller $R_{angular}$. We can find that the high requirement of spectral information will have cost in angular resolution.

5. Bottleneck of the resolution tradeoffs

The fundamental limitation of SLIM system is the spatial bandwidth product. Enlarge the sensor size allowing the entire system to receive more light, which raises the pixel numbers enabling the entire system of more detailed sampling. Therefore, the resolution of all four dimensions ($x + y + z + \lambda$) will be improved.

Through the above analysis, to give a clear and simple conclusion, the tradeoff relationship is shown in the following table:

Trade-off Table				
	In-plane Spatial resolution	Spectral resolution	depth resolution	Angular resolution
In-plane Spatial Resolution		$R_{spectral} + R_{spatial} = \xi \frac{D_{lens}}{P}$	$R_{depth} = \frac{\xi D_{lens}}{2P}$	$R_{angular} = \frac{1}{\frac{P(\xi \frac{D_{lens}}{P} - R_{spatial})}{w_0} - 1}$
Spectral resolution	$R_{spectral} + R_{spatial} = \xi \frac{D_{lens}}{P}$		$\eta(R_{spectral} + R_{spatial}) + 2(1 - \eta)R_{depth} = \xi \frac{D_{lens}}{P}$	$R_{angular} = \frac{1}{\frac{PR_{spectral}}{w_0} - 1}$
Depth resolution	$R_{depth} = \frac{\xi D_{lens}}{2P}$	$\eta(R_{spectral} + R_{spatial}) + 2(1 - \eta)R_{depth} = \xi \frac{D_{lens}}{P}$		$R_{angular} = \frac{1}{\frac{nPR_{depth}}{\xi fNA} - 1}$
Angular resolution	$R_{spatial} = \xi \frac{D_{lens}}{P} - \frac{w_0(R_{angular} + 1)}{PR_{angular}}$	$R_{spectral} = \frac{w_0(R_{angular} + 1)}{PR_{angular}}$	$R_{depth} = \frac{\xi fNA(R_{angular} + 1)}{nPR_{angular}}$	

Figure R8: Tradeoff relationship table between in-plane spatial resolution, spectral resolution, depth resolution and angular resolution.

Description of tradeoffs between in-plane spatial resolution, depth resolution, spectral resolution, and angular resolution has been added to the Supplementary material as Section 6.

Reviewers' Comments:

Reviewer #3:

Remarks to the Author:

The manuscript is now vastly improved over the original, and contains the key information needed to understand the optical image formation, reconstruction, and various trade-offs. The paper is now acceptable for publication.